# Patient-reported outcomes evaluation and assessment of facilitators and barriers to physical activity in the Transplantoux aerobic exercise intervention

Evi Masschelein[1‡], Stefan De Smet[2,3‡], Kris Denhaerynck[4], Laurens J. Ceulemans[5,6], Diethard Monbaliu[2,3,7], Sabina De Geest[4,8]*

1 Laboratory of Exercise and Health, Department of Health Sciences and Technology, ETH Zürich, Zurich, Switzerland, 2 Abdominal Transplantation, Department of Microbiology, Immunology and Transplantation, KU Leuven, Leuven, Belgium, 3 Transplantoux Foundation, Leuven, Belgium, 4 Institute of Nursing Science, Department Public Health, Faculty of Medicine, University of Basel, Basel, Switzerland, 5 Thoracic Surgery, University Hospitals Leuven, Leuven, Belgium, 6 Chronic Diseases and Metabolism, Laboratory of Respiratory Diseases and Thoracic Surgery (BREATHE), KU Leuven, Leuven, Belgium, 7 Department of Abdominal Transplant Surgery, University Hospitals Leuven, Leuven, Belgium, 8 Academic Centre for Nursing and Midwifery, Department of Public Health and Primary Care, KU Leuven, Leuven, Belgium

‡ EM and SDS contributed equally to this work and joint first authorship.
* sabina.degeest@kuleuven.be

**Data Availability Statement:** All relevant data are within the manuscript and its Supporting Information files.

## Abstract

### Background

Transplantoux's MVT exercise intervention prepares organ transplant recipients to cycle or hike up France's Mont Ventoux. We aimed to assess (i) MVT's effects on patient-reported outcomes (PROs) and (ii) perceived barriers and facilitators to physical activity.

### Methods

Using a hybrid design, a convenience sample of transplant recipients participating in MVT (n = 47 cycling (TxCYC); n = 18 hiking (TxHIK)), matched control transplant recipients (TxCON, n = 213), and healthy MVT participants (HCON, n = 91) completed surveys to assess physical activity (IPAQ), health-related quality of life (HRQOL; SF-36 and EuroQol VAS), mental health (GHQ-12), and depressive symptomatology, anxiety, and stress (DASS-21) at baseline, then after 3, 6 (Mont Ventoux climb), 9, and 12 months. TxCYC and TxHIK participated in a 6-month intervention of individualized home-based cycling/hiking exercise and a series of supervised group training sessions. Barriers and facilitators to physical activity (Barriers and Motivators Questionnaire) were measured at 12 months.

### Results

Regarding PROs, except for reducing TxHIK stress levels, MVT induced no substantial intervention effects. For both TxCYC and TxHIK, between-group comparisons at baseline showed that physical activity, HRQOL, mental health, depressive symptomatology and stress were similar to those of HCON. In contrast, compared to TxCYC, TxHIK, and HCON,

**Funding:** DM is a senior clinical investigator of the Flanders, Belgium research foundation. This research was made possible thanks to the financial support of the government of Flanders (EBO EG 101 3300). EM and SDS were funded by the Transplantoux Foundation, Leuven, Belgium. The funders had no role in study design, data collection and analysis, decision to publish, or preparation of the manuscript.

**Competing interests:** The authors have declared that no competing interests exist.

**Abbreviations:** BP, body pain; DASS, Depression, Anxiety and Stress Scale; GH, general health; GHQ, General Health Questionnaire; HCON, healthy controls participating in the Transplantoux program; HRQOL, health-related quality of life; IPAQ, International Physical Activity Questionnaires; MCS, mental component score; MET, metabolic equivalent of task; MH, mental health; PCS, physical component score; PF, physical functioning; PROs, patient-reported outcomes; RE, role limitations due to emotional health; RP, role limitations owing to physical health; SF, social functioning; SF-36, 36-item Short Form Health Survey; TxCON, control transplant recipients; TxCYC, transplant recipients participating in the Transplantoux cycling program; TxHIK, transplant recipients participating in the Transplantoux hiking program; VAS, visual analogue scale; VT, vitality.

physical activity, HRQOL and mental health were lower in TxCON. TxCON also reported greater barriers, lower facilitators, and different priority rankings concerning physical activity barriers and facilitators.

## Conclusion

Barely any of the PROs assessed in the present study responded to Transplantoux's MVT exercise intervention. TxCON reported distinct and unfavorable profiles regarding PROs and barriers and facilitators to physical activity. These findings can assist tailored physical activity intervention development.

## Trial registration

**Clinical trial notation:** The study was approved by the University Hospitals Leuven's Institutional Review Board (B322201523602).

## Introduction

In recent decades, advances in solid organ transplantation have contributed to improvements in short-term postoperative survival [1–5]. However, improvement in long-term survival remains a major challenge [2, 4, 5]. As increasing evidence shows the benefits of posttransplant exercise interventions on cardiorespiratory fitness [6–10], muscle strength [7–11], metabolic health [12], and quality of life [7–9], it is increasingly recognized that implementation of physical activity, possibly including exercise training, is an essential component of the posttransplant regimen. Still, despite empirical evidence and clinical recommendations [13–17] highlighting the need for physical activity and exercise, transplant recipients' adherence to regular physical activity and/or exercise is low for a multitude of reasons [18–22].

In 2008, to redress this deficiency, the University Hospitals Leuven (UZ Leuven) initiated Transplantoux (Transplantation + Mont Ventoux (MVT)), an organization run by patients, health care providers, and other stakeholders to promote physical activity in transplant recipients' daily life [23]. Transplantoux's MVT program includes a 6-month individualized moderate- to high-intensity exercise program that prepares its participants to cycle or hike up Mont Ventoux—a 25.9 km climb with a gentle (4%) average slope.

With its increasing popularity, Transplantoux has expanded to include transplant programs in Flanders (Belgium) and the Netherlands. In addition to MVT, it now also offers a wide range of opportunities for transplant recipients, living donors, and their relatives to engage in low-threshold physical activity throughout the year. Regarding clinical cardiorespiratory fitness outcomes in stable transplant recipients, MVT's safety and success have been shown elsewhere [23, 24]; however, MVT has not been evaluated from the perspective of patient-reported outcomes (PROs).

PROs, *i.e.*, outcomes collected directly from the patient without interpretation by clinicians or others, are proving increasingly useful to gain a more complete perspective of intervention effects: in addition to, for instance, clinical or physiological outcomes, they convey patients' unique, subjective views or experiences [25].

Exercise training interventions in transplant recipients have been evaluated in view of PROs such as health-related quality of life (HRQOL), depressive symptomatology, perceived

stress, and anxiety [26–28]. And while PROs reflect subjective concepts, they are assessed using standardized measures [29–32].

Two excellent examples of PROs are perceived barriers and facilitators to physical activity, data which can shed light on underlying factors that hinder or enhance engagement in regular physical activity or exercise training. Thus far, only a small number of studies have assessed perceived barriers and facilitators to physical activity following transplantation [33–36]. As their findings support the development and/or refinement of behavioral interventions, such explorations can inform further refinement of Transplantoux's physical activity interventions.

With this benefit in mind, the present study had two aims: (i) to assess and compare self-reported physical activity, HRQOL, mental health, depressive symptomatology, anxiety, and stress within and between transplant recipients participating in MVT (cycling or hiking), a matched control sample of non-participating transplant recipients, and healthy MVT participants over a 12-month period; and (ii) to assess and compare perceived barriers and facilitators to physical activity within these groups.

## Methods

### Study sample, design, and setting

The study protocol was previously published elsewhere (ClinicalTrials.gov identifier: NCT02533245) and conducted as planned. Four groups participated in this study: (1) a convenience sample of transplant recipients participating in the MVT cycling program (TxCYC); (2) a convenience sample of transplant recipients participating in the MVT hiking program (TxHIK); (3) a matched control group of transplant recipients not participating in the MVT program (TxCON); and (4) a convenience sample of healthy MVT participants (HCON) (**Fig 1**). Members of the TxCON group were matched using propensity score matching on 4 parameters (see below). All study participation was fully voluntary. The study was approved by the University Hospital Leuven's Institutional Review Board (B322201523602). Written informed consent was provided by every participant.

Inclusion criteria for transplant recipients participating in MVT were participating as a cyclist or hiker in MVT, ≥18 years of age, ≥6 months' transplant vintage, no episodes of acute rejection in the preceding 6 months, and Dutch language proficiency. Patients with cardiovascular or orthopedic contraindications regarding intensive physical exercise were excluded from participation. Patients from the TxCON group were selected from the UZ Leuven transplant database. Using uni- and multivariable logistic regression model analysis, a propensity score was estimated for all patients. Variables included in the propensity model were based on existing literature and included type of organ transplant, age, gender, and transplant vintage. Matching of these propensity scores did not result in an exact match for all parameters and therefore a multiple imputation was performed, resulting in different n (n = 3–4 matched TxCON to n = 1 transplant recipient participating in MVT). If more than 4 possible candidate TxCON patients were available, then a random selection of 4 was made. Inclusion criteria for TxCON were non-participation in MVT and Dutch proficiency. The HCON convenience sample was recruited from health care providers, family members, and friends of transplant recipients participating in the MVT program. Exclusion criteria for this group included cardiovascular, orthopedic, or any other contraindications regarding intensive physical exercise, or lack of Dutch proficiency.

Our hybrid study design combined (i) a longitudinal quasi-experimental design for longitudinal follow-up of physical activity, health-related quality of life, mental health, depressive symptomatology, anxiety, and stress, and (ii) a comparative cross-sectional design to compare barriers and facilitators to physical activity at the end of follow-up (**Fig 2**).

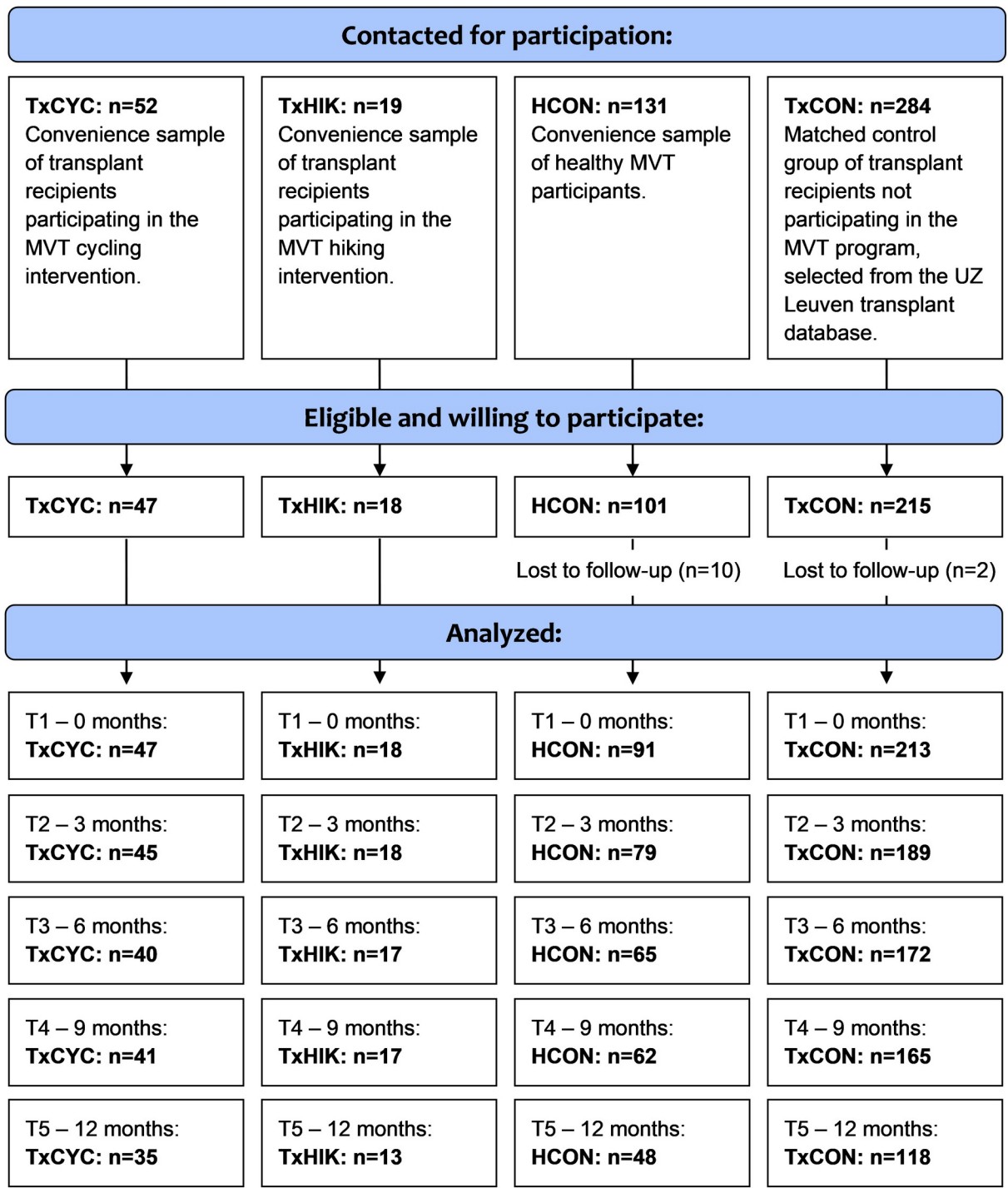

**Fig 1. CONSORT flowchart adjusted for non-randomized trial design.** All individuals who completed the survey at baseline (T1) were included in the present study. Study participants who did not to fill out one or more follow-up assessments were not excluded from the study, but were considered to have missed measurements for those time points.

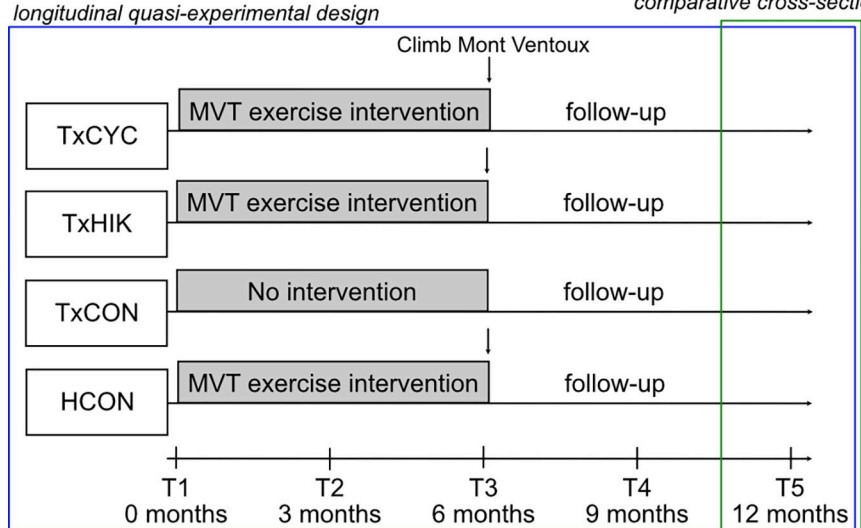

**Fig 2. Hybrid study design.** The present study combines a quasi-experimental design for longitudinal follow-up of physical activity, health-related quality of life, mental health, depressive symptomatology, anxiety, and stress with a cross-sectional design to compare barriers and facilitators to physical activity.

## Intervention

TxCYC and TxHIK participated in a 6-month exercise training program consisting of individualized home-based and self-monitored cycling or hiking training sessions and a series of supervised group cycling or hiking sessions. These patients underwent a cardiopulmonary exercise test to determine their baseline cardiorespiratory fitness and heart rate training zones. For each transplant recipient participating in MVT, a certified physiotherapist or physical trainer incorporated successful elements of existing exercise programs into an MVT regimen matched to the patient's preferences, abilities, and baseline cardiopulmonary exercise test results. The same physiotherapist or physical trainer subsequently coached that patient throughout the program.

TxCYC cycled at least three times per week for a minimum of 30 minutes per session, while self-monitoring their heart rate via a continuous heart rate monitor. In phase 1 (months 1–2), participants exercised at low to moderate relative intensity (60–75% of individual peak heart rate). In phase 2 (months 3–4), the training volume was increased to a series of short intervals of up to four minutes at moderate to vigorous relative intensity (75–85% of individual peak heart rate). In phase 3 (months 5–6), the training volume was maintained but focused predominantly on series of vigorous-intensity intervals of up to two minutes (85–90% of individual peak heart rate) as well as uphill cycling. When appropriate, programs were adjusted to match participants' individual needs, abilities, and functional limitations. Physical and/or psychological issues were discussed and managed accordingly, and encouragement was given. TxHIK participants received instructions regarding the training volume (time and distance) and speed (low, moderate, or high). Patients were instructed to hike three times a week: two short sessions of 30 to 45 min in duration and one longer session starting at 2 hours duration in month 1 and increasing to 8 hours' duration in month 6. TxCYC and TxHIK group members received their home-based training schedules and discussed their training progress on a monthly basis, normally by e-mail. From the third month onwards, transplant recipients in both the cycling and the hiking group participated in biweekly group training sessions designed and supervised by their coaches. A total of 7 such group sessions were offered.

The HCON exercise intervention was neither as controlled nor as closely supervised as those for the TxCYC and TxHIK groups; nor did its members receive home-based personalized training. However, participants were encouraged to train at least 3 times per week and to participate in the supervised group training sessions.

## Variables and measurement

Subject characteristics—age, gender, marital status, educational level, height, weight and smoking status—were collected via a self-reported web-based questionnaire. Transplant type and date of transplantation were retrieved from the UZ Leuven transplant database.

**Patient-reported outcomes.** *Perceived physical activity* was assessed with the Short Form of the International Physical Activity Questionnaires (IPAQ). This questionnaire was developed to monitor physical activity in 18–65 year old adults in diverse settings [37]. The first six questions involve the time participants have been physically active over the past 7 days.

The continuous score was calculated as an index of metabolic equivalent of task minutes (MET-min) per week. One MET corresponds to the rate of energy expenditure while sitting at rest. By convention, this equals an oxygen uptake of 3.5 mL·kg$^{-1}$·min$^{-1}$. Weekly energy expenditure was expressed as 'MET-min per week' and calculated as the product of MET intensity multiplied by activity time over the last seven days. Following the IPAQ scoring protocol, vigorous or moderate activity and walking consume respectively 8.0, 4.0, and 3.3 METs [38]. The IPAQ short form's final question—regarding average sitting time on a weekday—was omitted in the present study. The IPAQ questionnaires produce repeatable data and criterion validity comparable to most other self-report physical activity measures [37].

Respondents are asked to record the number of days they have done (i) vigorous, (ii) moderate, and (iii) walking activities ≥10 min in duration, as well as the time they spent doing such activities on each of those days. Based on the answers, both categorical (low, moderately, or highly active) and continuous scores are calculated. Criteria for moderately active categorization include: (i) ≥3 days of vigorous activity for at least 20 minutes per day; or (ii) ≥5 days of moderate-intensity activity and/or walking for at least 30 minutes per day; or (iii) ≥5 days of any combination of walking, moderate-intensity, or vigorous-intensity activities achieving ≥600 MET-min per week. Criteria for categorization as highly active include: (i) vigorous activity over ≥3 days, accumulating ≥1500 MET-min per week; or (ii) any combination of walking, moderate-intensity, or vigorous-intensity activities over ≥7 consecutive days, achieving ≥3000 MET-min per week.

*Perceived barriers and facilitators to physical activity* were measured at the final assessment timepoint (**Fig 2**) using a self-report instrument, the Barriers and Motivators Questionnaire. Listing 31 barriers and 23 facilitators on a 4-point scale (1 = 'not at all', 2 = 'slightly', 3 = 'moderately', 4 = 'very much'), this questionnaire was originally developed for hemodialysis patients [39] but has also been used in solid organ transplant recipients [18, 31, 36]. An unrotated principal component analysis on the barrier instrument showed that 48% of the indicated variability was shared among all items, with a Cronbach's alpha of 0.92. Facilitator items shared 42% of all variability, with an alpha of 0.91. This allowed us to calculate the arithmetic mean scores over all items per scale. Percentages of positive answers on individual items were calculated based on responses of 'slightly' or higher.

*Health-related quality of life* was assessed via the 36-item Short Form Health Survey (SF-36). This self-report instrument is a general health status instrument that measures limitations resulting from poor health and/or bodily pain on eight categories of physical, social, and role activities: physical functioning (PF), role limitations owing to physical health (RP), body pain (BP), general health (GH), vitality (VT), social functioning (SF), role limitations due to

emotional health (RE), and mental health (MH). These domains are summarized into a physical component score (PCS) and a mental component score (MCS). Possible scores range from 0–100, with higher scores being more positive (*i.e.*, less pain, less limitation). The PCS was calculated as the weighted sum of standardized scale scores from the PF, RP, BP, and GH scales. The MCS was calculated from the VT, SF, RE, and MH scales. The SF-36 has been validated in both general and transplant populations [29, 40].

HRQOL was also assessed via the visual analogue scale (VAS) developed by the Euro-QolGroup [41, 42]. Patients self-rated their health using a vertical rating scale with 'best imaginable health state' (value = 100) at the top and 'worst imaginable health state' (value = 0) at the bottom. The EuroQol VAS shows responsiveness to quality of life improvements from before to after transplantation [43].

Self-reported *depressive symptomatology*, *anxiety*, *and stress* were measured via the Depression, Anxiety and Stress Scale (DASS-21), a short version of the 42-item DASS [44]. Patients were asked to score each item for the past week on a 4-point Likert scale. Response options ranged from 0 (did not apply to me at all) to 3 (applied to me very much or most of the time). The intensity of each of the three conditions was determined by summing the scores of responses to its 7-item subscale, with higher scores reflecting higher symptom severity. To convert the results to full-scale DASS scores, each total (range 0–21) was doubled. The subscales of the DASS-21 and the DASS-42 have good reliability (internal consistency, temporal stability, factor structure) and convergent/divergent validity both in general and in clinical populations [45–48].

*Mental health* was scored via the 12-item, unidimensional General Health Questionnaire (GHQ–12). The GHQ-12 is a screening tool used to identify the severity of psychological distress experienced by an individual over the past few weeks. For each item, respondents indicate whether they have recently experienced a particular symptom or behavior. Items are rated on a four-point scale (less than usual, no more than usual, rather more than usual, or much more than usual), to which a bimodal scoring method (with respective values of 0-0-1-1) is applied. The overall score, which is determined by summing all items' values, ranges from 0–12. A score of ≥4 indicates a high stress level. This instrument has been widely used. While it shows some tendency towards correlated errors among its negative items in the general population, it is considered a valid tool for research [49, 50].

## Data collection

Data were collected from January 2015 until January 2016. Assessment of PROs occurred at five time points (**Fig 2**): at baseline (T1), after 3 months (T2), after 6 months (subjects received the survey one week after the climb of Mont Ventoux, T3), after 9 months (3 months into follow-up, T4), and after 12 months (6 months into follow-up, T5). Telephone numbers and e-mail addresses of MVT participants (TxCYC, TxHIK, and HCON) and TxCON were retrieved from the Transplantoux organization and the University Hospital Leuven database, respectively, observing applicable data protection rules at the time of the study. All study candidates received a short call about the purpose of the study. Thereafter, those indicating an interest in participating received an email with background information, an invitation to participate in the study, and a link to the online survey. The online survey was completed via a web-based survey tool provided by KU Leuven (http://websurvey.kuleuven.be).

All subjects signed an online informed consent form. Each survey had a unique code that allowed identification of the subject by the investigator. Individuals who did not fill in the questionnaire after 2 weeks received a reminder. If they had not returned the questionnaire 7 days after the first reminder, a second reminder was sent. If they still had not returned the

questionnaire 14 days after the second reminder, they received a reminder telephone call (for baseline measurement only). Procedures were repeated at each data collection time point, but not for the informed consent procedure. All individuals who completed the survey at baseline were included in the present study. Study participants who failed to complete one or more follow-up assessments were not excluded from the study.

### Data analysis

Descriptive statistical measures of central tendency and dispersion were used as appropriate. For normally distributed PROs, general linear mixed regression analysis was chosen; for non-normally distributed PROs, generalized linear mixed regression analyses were applied. Specifically, an ordinal logit model was chosen. The random-intercepts approach allowed the retention of patients who missed measurements during follow-up. For all of these models, 'patient' was entered as a random-intercepts variable; time, study group, and the interaction of time x group were entered as fixed covariates. Individual study groups' regression slopes were tested if significantly different from zero, and contrasted with one another. Contrasts indicate the locations of (possibly significant) interaction effects. For each outcome, two analyses were performed—the first to test the effect of the exercise intervention (T1 to T3), the second to test the sustainability of the training effects (T3 to T5). General(ized) linear regression analysis was also used to perform post-hoc testing for possible baseline differences in outcome variables between the study groups.

Barriers and facilitators were analyzed at the levels of their respective average total scores and of individual item scores. In the first case, total scores between study groups were compared using a linear model. In the latter, the six top-ranked barriers and facilitators were ranked using mean scores over all patients and percentages of positive answers. Analyses were executed using SAS 9.4 (SAS Institute, Cary, NC). P<0.05 was considered statistically significant.

## Results

### Participants' demographic characteristics

**Fig 1** shows the sample flowchart for all groups. At baseline (T1) the sample included 65 transplant recipients participating in MVT (TxCYC: n = 47; TxHIK: n = 18), 213 TxCON, and 91 HCON. **Table 1** summarizes these groups' demographic characteristics. The lowest survey response rates—ranging from 53%–74% depending on the group—occurred at the final follow-up assessment (T5) (**Fig 1**).

### Group differences in PROs at baseline

Between-group baseline differences in PROs are presented in **Figs 3–5** and **S1 and S2 Tables**. These indicated that self-reported physical activity (continuous and categorical outcomes), HRQOL (SF-36 and EuroQol VAS), mental health, depressive symptomatology and stress in both TxCYC and TxHIK were similar to those in HCON. Compared to TxCON, TxCYC scored significantly higher self-reported physical activity (continuous and categorical outcomes), HRQOL (SF-36 and EuroQol VAS), and mental health and reported significantly lower depressive symptomatology, anxiety, and stress. Compared to TxCON, TxHIK scored significantly higher self-reported physical activity (categorical outcome) and HRQOL (SF-36 and EuroQol VAS) and tended (P = 0.07) to score better for mental health. Depressive symptomatology, anxiety, and stress in TxHIK did not differ from those of TxCON. Unsurprisingly, HCON scored significantly higher regarding physical activity (categorical outcome), HRQOL

**Table 1. Demographic characteristics of the study population at baseline.**

| Variable | Values | TxCYC (n = 47) | TxHIK (n = 18) | TxCON (n = 213) | HCON (n = 91) |
|---|---|---|---|---|---|
| Age | Mean ± SD | 47.9 ± 11.2 | 54.2 ± 11.5 | 50.1 ± 11.5 | 49.5 ± 11.0 |
| Sex | Male–n (%) | 40 (85%) | 13 (72%) | 176 (83%) | 46 (51%) |
| Marital status | Single–n (%) | 14 (30%) | 2 (11%) | 49 (23%) | 11 (12%) |
| | Married/cohabiting–n (%) | 30 (64%) | 14 (78%) | 148 (69%) | 71 (78%) |
| | Divorced–n (%) | 3 (6%) | 2 (11%) | 13 (6%) | 9 (10%) |
| | Widow(er)–n (%) | 0 (0%) | 0 (0%) | 3 (1%) | 0 (0%) |
| Education | Primary–n (%) | 3 (6%) | 2 (11%) | 34 (16%) | 3 (3%) |
| | Secondary–n (%) | 11 (23%) | 2 (11%) | 76 (36%) | 14 (15%) |
| | Vocational/bachelor–n (%) | 27 (57%) | 10 (56%) | 82 (39%) | 49 (54%) |
| | Master or PhD–n (%) | 6 (13%) | 4 (22%) | 20 (9%) | 25 (27%) |
| BMI | Mean ± SD | 24.3 ± 4.1 | 25.1 ± 5.2 | 25.1 ± 4.6 | 24.4 ± 3.5 |
| Years since transplantation | Median (IQR) | 3.9 (2.3–9.4) | 3.04 (2.3–7.4) | 4.95 (2.4–8.9) | NA |
| Primary transplant | Heart–n (%) | 9 (19%) | 6 (33%) | 61 (29%) | NA |
| | Kidney–n (%) | 18 (38%) | 3 (17%) | 64 (30%) | NA |
| | Liver–n (%) | 11 (23%) | 3 (17%) | 50 (24%) | NA |
| | Lung–n (%) | 8 (17%) | 5 (28%) | 36 (17%) | NA |
| | Small bowel–n (%) | 1 (2%) | 0 (0%) | 1 (0%) | NA |
| | Stem cell–n (%) | 0 (0%) | 1 (6%) | 1 (0%) | NA |
| Smoking | Current–n (%) | 0 (0%) | 1 (6%) | 10 (5%) | 2 (2%) |
| | Past–n (%) | 16 (34%) | 9 (53%) | 83 (38%) | 27 (25%) |
| | Never–n (%) | 31 (66%) | 8 (47%) | 120 (55%) | 62 (56%) |
| | Packyears–median (IQR) | 9.0 (3.6–15.7) | 24.0 (10.8–30.0) | 17.0 (6.5–27.0) | 8.0 (2.0–14.0) |

HCON: healthy participants; NA: not applicable; TxCON: control transplant recipients; TxCYC: transplant recipients participating in the cycling program; TxHIK: transplant recipients participating in the hiking program

(SF-36 and EuroQol VAS), and mental health than TxCON, and reported significantly lower depressive symptomatology and anxiety. However, stress levels did not differ between HCON and TxCON.

## Effect of the MVT training intervention on PROs

**Figs 3–5** illustrate the evolution of all study groups' PROs. Regression slopes from baseline (T1) to 6 months (T3, ~1 week after summiting Mont Ventoux) and from 6 months to 12

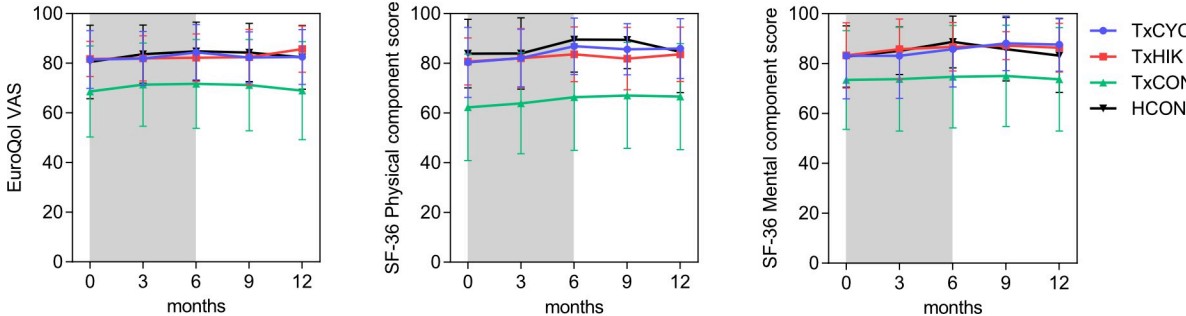

**Fig 3. Health-related quality of life for all the groups at each time point using the EuroQol VAS and SF-36 questionnaire.** Line graphs are mean ± SD. The grey bar indicates the Transplantoux intervention.

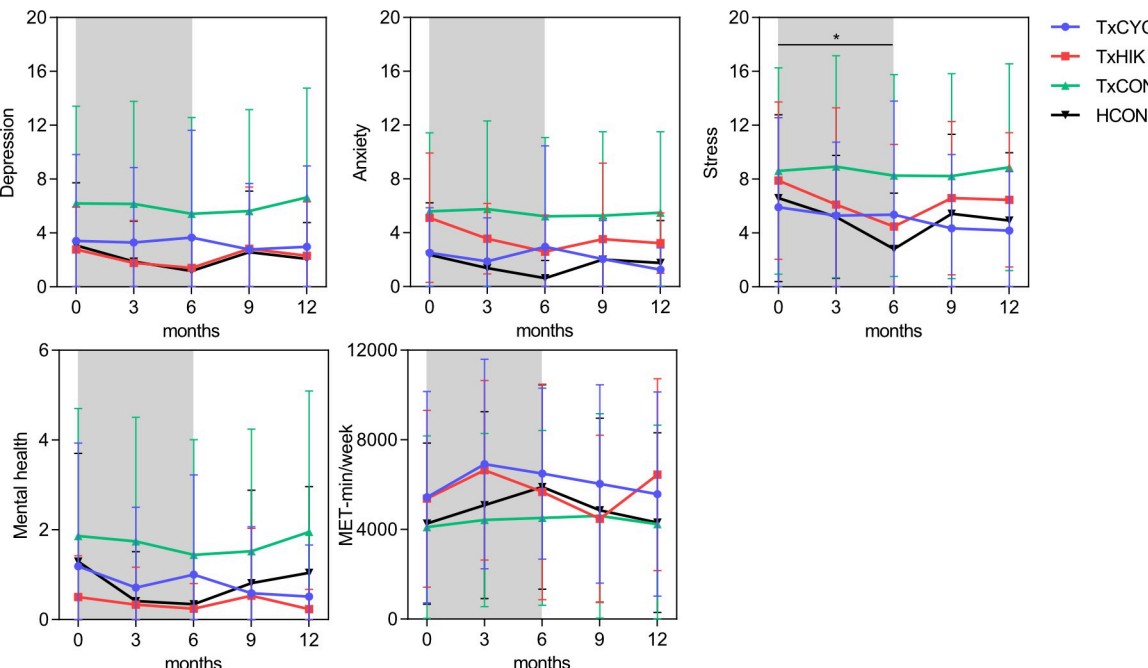

**Fig 4. Depression, anxiety, stress, mental health, and physical activity (continuous score) for all groups at each time point.** Line graphs are mean ± SD. Grey bar indicates the Transplantoux intervention. *P<0.05 group x time interaction effect.

months (T5) show within-group time effects in TxCYC, TxHIK, HCON, and TxCON (**S3 and S4 Tables**). In TxCON, self-reported levels of physical activity, HRQOL, mental health, depressive symptomatology, and anxiety slightly though significantly improved before and/or deteriorated after the summer (T3), *i.e.*, seasonal effects were evident.

To test the training intervention's effects, we performed regression slope contrast analyses of all PROs. In view of physical activity, HRQOL, mental health, and depressive

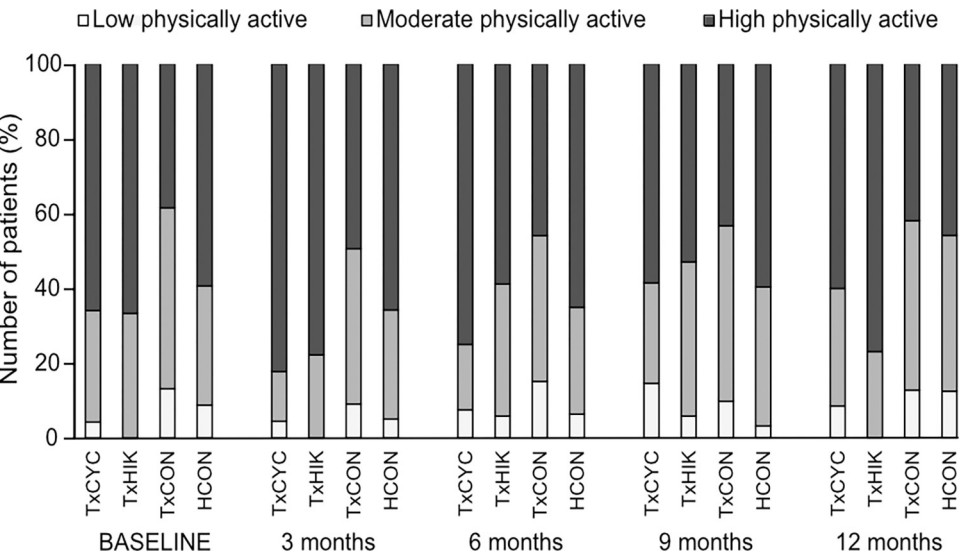

**Fig 5. Physical activity status (categorical score) for all groups at each time point.**

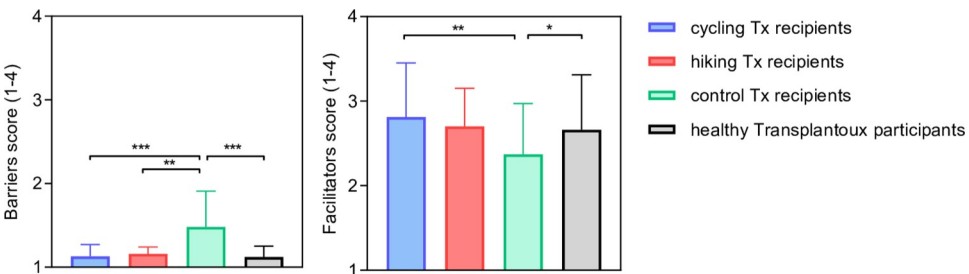

**Fig 6. Barriers and facilitators to physical activity for each group at the end of the study.** Bar graphs are mean ± SD. ***P<0.001, **P<0.01, *P<0.05.

symptomatology, no intervention effects were observed in TxCYC or TxHIK compared to TxCON. A significant intervention effect was initially found regarding stress scores, which decreased in TxHIK (P<0.05) but remained unchanged in TxCON. However, during follow-up (T3 to T5), TxHIK stress scores returned to levels approaching those of TxCON (P = 0.054), undoing most of the intervention effects.

## Comparison of barriers and facilitators to physical activity at the end of the study

Fig 6 shows all groups' ratings of barriers and facilitators. The average barrier scores did not differ between TxCYC (1.13 ± 0.14; p<0.001), TxHIK (1.16 ± 0.08; P<0.01), and HCON (1.12 ± 0.13; P<0.001), but were significantly lower in these groups than in TxCON (1.47 ± 0.42). The average facilitator scores of TxCYC (2.81 ± 0.64; P<0.01) and HCON (2.66 ± 0.65, P<0.05) were significantly higher than those of TxCON (2.37 ± 0.59). No differences in facilitator scores were found between TxCYC, TxHIK, or HCON. Table 2 lists the six highest scoring barriers and facilitators for each group. The three highest ranked barriers to physical activity were identical for TxCYC, TxHIK, and HCON (1. bad weather, 2. lack of time, 3. too fatigued). These differed from TxCON, who reported unpleasant sensations associated with exercise as their primary barrier to physical activity, followed by being too fatigued and lack of motivation. Fifty-six percent of TxCON *versus* negligible numbers of TxCYC and TxHIK acknowledged the barrier 'low expectations of self to exercise.' Being too fatigued and lack of motivation were acknowledged respectively within top 3 and top 5 barriers by all study groups.

Rankings of facilitators were more similar, with 'feeling healthy' ranked first by all groups. 'Enjoying how exercise feels' was listed as the second or third most important facilitator in TxCYC, TxHIK, and HCON, but was not acknowledged as a top-6 facilitator in TxCON. 'Belief in one's ability to be physically active' was a recurrent facilitator in TxCYC and TxHIK, but not in TxCON. In its place, TxCON favored 'wanting to manage weight'.

## Discussion

As PROs assess intervention outcomes directly from the patient perspective, they are increasingly used to evaluate the effects of physical exercise interventions [26]. The MVT intervention, a 6-month post-transplantation exercise training program culminating in an ascent of Mont Ventoux either by bicycle (TxCYC) or on foot (TxHIK), led to temporarily decreased stress in TxHIK but failed to impact other PROs. However, transplant recipients motivated to engage in this exercise intervention reported quality of life, depressive symptomatology, stress, and mental health ratings similar to those of the healthy control (HCON) group. In contrast,

**Table 2. First six barriers and facilitators to physical activity in each of the study groups.**

| Instrument | Study group | Item | Mean ± SD | N (%)* |
|---|---|---|---|---|
| Barriers | TxCYC | Bad weather | 1.74 ± 0.83 | 18 (53) |
| | | Lack of time | 1.65 ± 0.77 | 18 (53) |
| | | Being too fatigued | 1.31 ± 0.53 | 10 (29) |
| | | Fear of injury | 1.26 ± 0.44 | 9 (26) |
| | | Lack of motivation | 1.26 ± 0.51 | 8 (23) |
| | | Co-morbid health problems | 1.21 ± 0.48 | 6 (18) |
| | TxHIK | Bad weather | 1.83 ± 0.83 | 7 (58) |
| | | Lack of time | 1.50 ± 0.52 | 6 (50) |
| | | Being too fatigued | 1.46 ± 0.52 | 6 (46) |
| | | Lack of motivation | 1.31 ± 0.48 | 4 (31) |
| | | Unpleasant sensations associated with exercise | 1.31 ± 0.48 | 4 (31) |
| | | Low expectations by health care providers | 1.31 ± 0.63 | 3 (23) |
| | TxCON | Unpleasant sensations associated with exercise | 1.99 ± 0.96 | 72 (62) |
| | | Being too fatigued | 1.97 ± 0.97 | 71 (60) |
| | | Lack of motivation | 1.83 ± 0.91 | 66 (56) |
| | | Preferring to spend time doing other things | 1.79 ± 0.88 | 62 (53) |
| | | Low expectations by self to exercise | 1.77 ± 0.83 | 66 (56) |
| | | Bad weather | 1.74 ± 1.00 | 50 (45) |
| | HCON | Bad weather | 1.65 ± 0.74 | 23 (50) |
| | | Lack of time | 1.56 ± 0.77 | 21 (44) |
| | | Being too fatigued | 1.46 ± 0.65 | 19 (40) |
| | | Preferring to spend time doing other things | 1.28 ± 0.50 | 12 (26) |
| | | Lack of motivation | 1.27 ± 0.57 | 10 (21) |
| | | Fear of injury | 1.19 ± 0.44 | 9 (19) |
| Facilitators | TxCYC | Feeling healthy | 3.71 ± 0.62 | 35 (100) |
| | | Enjoying how exercise feels | 3.60 ± 0.69 | 35 (100) |
| | | Belief in one's ability to be physically active | 3.49 ± 0.70 | 35 (100) |
| | | Wanting increased health | 3.46 ± 0.82 | 35 (100) |
| | | Wanting increased strength | 3.43 ± 0.78 | 35 (100) |
| | | Wanting increased energy | 3.40 ± 0.91 | 34 (97) |
| | TxHIK | Feeling healthy | 3.54 ± 0.78 | 13 (100) |
| | | Exercising with others | 3.23 ± 0.93 | 13 (100) |
| | | Enjoying how exercise feels | 3.23 ± 0.93 | 12 (92) |
| | | Wanting increased energy | 3.15 ± 0.69 | 13 (100) |
| | | Knowing the value of increased exercise | 3.15 ± 0.99 | 12 (92) |
| | | Belief in one's ability to be physically active | 3.15 ± 0.99 | 12 (92) |
| | TxCON | Feeling healthy | 3.38 ± 0.86 | 114 (97) |
| | | Wanting increased health | 3.05 ± 0.93 | 112 (97) |
| | | Wanting increased energy | 2.98 ± 1.06 | 102 (89) |
| | | Wanting to manage weight | 2.96 ± 1.05 | 105 (90) |
| | | Wanting increased strength | 2.88 ± 0.98 | 107 (91) |
| | | Knowing the value of increased exercise | 2.86 ± 0.99 | 105 (90) |
| | HCON | Feeling healthy | 3.60 ± 0.79 | 46 (96) |
| | | Enjoying how exercise feels | 3.38 ± 0.91 | 46 (96) |
| | | Knowing the value of increased exercise | 3.31 ± 0.99 | 44 (92) |
| | | Wanting increased health | 3.31 ± 1.01 | 43 (90) |
| | | Wanting to feel better | 3.26 ± 1.03 | 43 (91) |

(*Continued*)

**Table 2.** (Continued)

| Instrument | Study group | Item | Mean ± SD | N (%)* |
|---|---|---|---|---|
| | | Wanting increased energy | 3.25 ± 0.89 | 45 (94) |

*Number (and percentage) of subjects indicating that at least 'slightly' pertained to an item. HCON: healthy participants; NA: not applicable; TxCON: control transplant recipients; TxCYC: transplant recipients participating in the cycling program; TxHIK: transplant recipients participating in the hiking program

the transplant control (TxCON) group's self-ratings for physical activity, quality of life, and mental health were all lower than any those of any other group. Lastly, the present study identified specific barriers and motivators to physical activity in TxCON as compared to TxCYC, TxHIK, and HCON. These insights provide a crucial building block for intervention development and choice of relevant implementation strategies for these least active transplant recipients (TxCON) that could benefit most from getting more physically active. This area of research should get priority as currently these patients typically are not reached in RCTs nor in daily clinical practice.

As has been shown by patients who received the MVT intervention, physical activity results in improved physiological outcomes [23, 24], which we would expect to result in similarly improved PROs. In our study this is suggested by the major difference between control patients and those participating in the MVT intervention.

While our group's research in transplant recipients participating in MVT has shown intervention effects on physiological parameters such as cardiorespiratory fitness [23, 24], no perceived increase in physical activity was reported ~1 week after summiting Mont Ventoux. This could reflect a decompression effect following the attainment of a major personal challenge.

Meanwhile, other PROs, which in MVT participants were already quite good at baseline, remained stable compared to TxCON. Indeed, these participants already showed PROs similar to those of the healthy controls at baseline. Lack of sensitivity of instruments used in transplantation to pick up an intervention effect could be involved. In addition, substantial variability was observed in PROs at and between the assessment time points. In TxCON, self-reported levels of physical activity, HRQOL, mental health, depressive symptomatology, and anxiety slightly but significantly improved before and/or deteriorated after the summer (T3). This observation suggests that seasonal effects described elsewhere may have induced noise in our data, potentially hampering our ability to tease out intervention effects [51–54]. However, it also indicates that bad weather hampers physical activity engagement and that there is room for improvement in PROs, at least in TxCON.

Indeed, both in RCTs and in observational studies, favorable effects of exercise training—including reduced fatigue, anxiety, and depressive symptomatology [55–58]—are commonly, though not consistently [6, 59–61] reported as examples of improved quality of life [7–10] in solid organ transplant recipients. To counter this tendency, future research that evaluates exercise interventions should assess physiological parameters and PROs simultaneously. Observing intervention effects both objectively and from a patient (PRO) perspective would add detail to the current picture, allowing evaluation of which parameters are most sensitive to change.

The matched TxCON PROs showed overall poorer baseline scores than either their TxCYC or their TxHIK counterparts. The TxCON PROs profile included lower physical activity, greater perceived barriers to physical activity, and lower ratings of facilitators to physical activity. Particularly compared to reports of healthy individuals and transplant recipients participating in MVT, such perceptions reflected not only this group's priority rankings of barriers and facilitators to physical activity, but possibly also their reported quality of life and mental health.

For example, whereas TxCYC, TxHIK, and HCON reported enjoying exercise-associated sensations, TxCON disliked them. TxCON reported higher barrier and lower facilitator scores than any other group. More than 60% of TxCON—compared to 31% in TxHIK and negligible numbers in TxCYC and HCON—reported that 'unpleasant sensations associated with exercise' restrained them from engaging in it. Similarly, regarding exercise facilitators, whilst TxCYC and TxHIK acknowledged the importance of 'enjoying how exercise feels' and 'belief in one's ability to be physically active', TxCON did not list either of these as priority facilitators. TxCON also differed from the other study groups in listing 'low expectations regarding exercise' as an important barrier.

Our findings are congruent with those of Sanchez *et al.*, who reported that, among their kidney transplant recipients, those who were most physically active endorsed fewer barriers and more facilitators to physical activity than their more sedentary counterparts [36]. Furthermore, inactive kidney transplant recipients less often counted 'enjoying how exercise feels', 'belief in one's ability to be physically active', 'wanting to feel better' and 'wanting increased energy' as relevant facilitators to physical activity [36]. The present study also corroborates Sanchez *et al.'s* assertion that lack of motivation, bad weather, and fatigue are important barriers to overcome. In contrast to our study, though, only 22% of their kidney transplant recipients indicated 'unpleasant sensations associated with exercise' as a barrier to exercise, while 83% reported 'enjoying how exercise feels' as a facilitator [36].

The distinct TxCON profile that emerged points to a need for alternative strategies to engage transplant recipients who are less physically active in physical activity programs. And while physical inactivity is highly prevalent among transplant recipients [18–22], sedentary and physically inactive patients would probably benefit the most from 'physical activity as medicine'.

Specific motivators and barriers identified in TxCON can support current physical activity initiatives, such as for instance organized by local patient organizations (*e.g.*, Transplantoux) or international federations (*e.g.*, World Transplant Games Federation) to more successfully target less or non-physically active patients. Especially developing adapted interventions and use of specific implementation strategies can support to reach these patients and to get and keep them more physically active. RCTs show a bias towards inclusion of more physically active patients or patients willing to engage in physical activity [62, 63]. Also in Transplantoux this has been the case, resulting in a selection bias. Sampling strategies should guarantee that the whole spectrum of transplant patients, ranging from not physically active at all to already physically active, are included in intervention studies. An approach tailored to the needs and preferences of the physically inactive transplant recipients should focus on finding strategies first to activate these patients, then to keep them physically active. Regarding this point, our TxCON's perceived barriers and facilitators offer a useful starting point [64, 65].

By shifting the focus from an optimal exercise dose to active living and enjoyment of physical activity, the idea of a physically active lifestyle can gain traction for sedentary and physically inactive transplant recipients and might stimulate active patients to persist in or even increase their levels of physical activity. To this end, interventions' attractiveness and acceptability could be enhanced by tailoring them to patients' preferences through co-design of the program.

One should consider that any increase in physical activity can reasonably be expected to result in clinical benefit [66, 67]. Reducing sedentary time in favor of physical activity of any intensity, including activities at light intensity, is now acknowledged and promoted as a path to health benefits [68]. Light-intensity physical activities that modestly improve patients' total physical activity level, whether or not they are implemented as steps toward more vigorous

and voluminous physical activity, could thus be offered to lower the threshold for physical activity initiation and maintenance whilst nonetheless generating positive clinical impacts.

Given the more pronounced exercise-related sensations associated with high-intensity interval training, it may seem somewhat counterintuitive that, at least in healthy and overweight populations, some versions of it may be recommended over moderate-intensity endurance training, as they tend to evoke more pronounced affective/enjoyment responses [69]. Furthermore, considering that lack of time is a commonly-named barrier, high- (compared to moderate-) intensity training has been suggested as a more time-efficient strategy to induce comparable or even superior health benefits [70, 71]. One significant principle is that lack of time for physical activity tends to indicate the allocation of a low priority to it in daily-life activities.

Therefore, integration and expansion of physical activity within daily-life practices that are not performed for enjoyment, but that improve physical fitness and health, may be useful. Examples of such 'incidental physical activity' include making one's daily commute on foot or by bicycle, carrying groceries, taking stairs, *etc.* A similar strategy might be effective to integrate key principles of high-intensity interval training into incidental physical activity patterns [72]. *I.e.*, short (0.5–5 min) sporadic incidental physical activities at vigorous intensity may include brisk walking, brisk bicycling, uphill walking, and taking stairs as means for transport to the bus stop, the supermarket, family visits, or work and could be part of daily-life with 3–5 of such activity bouts spread across a whole day.

Fatigue is a common symptom [73–79] and a well-known barrier to physical activity after transplantation [33, 35, 36]. At the same time, regular physical exercise is known to improve quality of sleep and symptoms of fatigue [55, 80–83]. Informing patients of the benefits of regular physical activity on symptoms of fatigue is therefore crucial.

The current study has certain notable limitations. First, we did not exclude patients who did not complete our survey during one or more of their follow-up assessments, leading to increasing variability in the study sample. Over time, the number of participants not completing the survey increased: at the intermediate (6 months) and final (12 months) measurement points, respectively 20% and 42% of patients did not fill out the survey. This may have introduced a selection bias to our findings, as the least motivated patients would be most likely to leave the study prematurely. Second, we did not objectively assess physical activity. Self-report measures are known to perform sub-optimally, typically overreporting physical activity levels [84]. In fact, this may account for our TxCON group's unexpectedly high reported activity levels. Clear between-group differences were found for categorical but not continuous indices of self-reported physical activity using IPAQ—a mismatch that may indicate an issue with IPAQ's terminology.

For example, vigorous physical activity is described as that requiring hard physical effort and making one breathe much harder than normal. As individuals with low cardiorespiratory fitness will breathe much more heavily at lower absolute exercise intensities and perceive lower absolute exercise intensity as hard effort, this definition would encourage them to define many moderate activities as vigorous.

Irrespective of the respondent's cardiorespiratory fitness, in calculating MET-min per week, vigorous activity is considered to consume 8 METs. In rough terms, stable transplant recipients present an average peak oxygen uptake between ~15 and 32 mL·kg$^{-1}$·min$^{-1}$, with thoracic and abdominal organ recipients situated respectively at the lower and higher ends of the spectrum [85]. This means that the majority of transplant recipients would be unable to exercise at 8 METs, the equivalent of 28 mL·kg$^{-1}$·min$^{-1}$. Therefore, it is entirely possible that the IPAQ definition led less active TxCON participants to overestimate their physical activity compared to their more physically fit TxCYC, TxHIK, and HCON counterparts.

One final limitation of the present study should also be considered. The intervention proto-col differed between transplant recipients and HCON: whereas transplant recipients received detailed training programs and follow-up prompts, HCON did not. However, our primary aim was to compare PROs in transplant recipients either participating or not participating in MVT.

## Conclusion

Regarding the measured PROs, a six-month dose of the MVT exercise training intervention lowered stress in the hiking transplant recipients, while other measurements remained stable. PROs in participating transplant recipients were similar to those of their healthy counterparts but more favorable than those of the matched control transplant recipients. In addition to being less physically active, control-group transplant recipients reported higher barriers and lower facilitators to physical activity and ranked these barriers and facilitators quite differently from the other three groups, all of which were quite similar to one another. This finding indi-cates that initiation and continuation of a physically active lifestyle in sedentary transplant recipients will require physical activity and exercise interventions tailored closely to their needs. To help future developers of physical activity interventions achieve this goal, we strongly recommend that they consider the barriers and facilitators identified here.

## Supporting information

**S1 Checklist. TREND statement checklist.**
(PDF)

**S1 Table. Patient reported outcomes for all groups on each timepoint.** HCON: healthy par-ticipants; TxCON: control transplant recipients; TxCYC: transplant recipients participating in the cycling program; TxHIK: transplant recipients participating in the hiking program.
(DOCX)

**S2 Table. Between-group differences of patient reported outcomes at baseline (T1).** * 1: cycling transplant recipients; 2: hiking transplant recipients; 3: control transplant recipients; 4: healthy participants.
(DOCX)

**S3 Table. Regression slopes per study group and slope contrasts of patient reported out-comes over time between baseline (T1) and intervention (T3, month 6).** * 1: cycling trans-plant recipients; 2: hiking transplant recipients; 3: control transplant recipients; 4: healthy participants.
(DOCX)

**S4 Table. Regression slopes per study group and slope contrasts of patient reported out-comes over time between intervention (T3, month 6) and study end (T5, month 12).** * 1: cycling transplant recipients; 2: hiking transplant recipients; 3: control transplant recipients; 4: healthy participants.
(DOCX)

**S1 Data.**
(CSV)

**S2 Data.**
(CSV)

**S1 Protocol.**
(PDF)

**S1 Dataset. Patient level data.**
(DOCX)

# Acknowledgments

The authors dedicate this manuscript to Mr. Raf Somers (†), one of the founders of Transplantoux, who co-designed the concept of this study. The authors would like to thank all volunteers for their enthusiastic participation in this study. We also acknowledge the KU Leuven Athletic Performance Center (Bakala Academy, KU Leuven, Leuven, Belgium) for their invaluable assistance with the training exercise programs and supervised group training sessions. We thank dr. Steffen Fieuws for his assistance in the statistical analyses. For logistical organization of the Mont Ventoux climb, we thank Sporta (Tongerlo, Belgium). And we thank professional medical writer Chris Shultis for his assistance in preparing the final version of the manuscript.

# Author Contributions

**Conceptualization:** Evi Masschelein, Laurens J. Ceulemans, Diethard Monbaliu, Sabina De Geest.

**Formal analysis:** Kris Denhaerynck.

**Investigation:** Evi Masschelein, Sabina De Geest.

**Methodology:** Evi Masschelein, Laurens J. Ceulemans, Diethard Monbaliu, Sabina De Geest.

**Project administration:** Evi Masschelein, Sabina De Geest.

**Visualization:** Evi Masschelein, Stefan De Smet, Kris Denhaerynck, Sabina De Geest.

**Writing – original draft:** Evi Masschelein, Stefan De Smet, Sabina De Geest.

**Writing – review & editing:** Evi Masschelein, Stefan De Smet, Kris Denhaerynck, Laurens J. Ceulemans, Diethard Monbaliu, Sabina De Geest.

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
