## [Decision Letter · Decision Letter 0]

13 May 2022

PONE-D-21-28527Patient-reported outcomes evaluation of the MVT exercise intervention and assessment of facilitators and barriers to physical activity in transplant recipientsPLOS ONE

Dear Dr. De Geest,

Thank you for submitting your manuscript to PLOS ONE. After careful consideration, we feel that it has merit but does not fully meet PLOS ONE’s publication criteria as it currently stands. Therefore, we invite you to submit a revised version of the manuscript that addresses the points raised during the review process.

Your manuscript has been assessed by two external experts, whose comments are appended below. The reviewers have raised some questions about the statistical analysis, and made some suggestions for additional discussion, among other points. Please ensure you respond to all of the issues raised by the reviewers in your response to reviewers document, and revise your manuscript accordingly. 

We look forward to receiving your revised manuscript.

Kind regards,

Joseph Donlan

Editorial Office

PLOS ONE

Journal Requirements:

4. Please upload a copy of Figure 6, to which you refer in your text on page 20. If the figure is no longer to be included as part of the submission please remove all reference to it within the text.

Reviewers' comments:

Reviewer's Responses to Questions

**Comments to the Author**

1. Is the manuscript technically sound, and do the data support the conclusions?

Reviewer #1: Partly

Reviewer #2: Yes

2. Has the statistical analysis been performed appropriately and rigorously? 

Reviewer #1: I Don't Know

Reviewer #2: Yes

3. Have the authors made all data underlying the findings in their manuscript fully available?

Reviewer #1: No

Reviewer #2: Yes

4. Is the manuscript presented in an intelligible fashion and written in standard English?

Reviewer #1: No

Reviewer #2: Yes

5. Review Comments to the Author

Reviewer #1: Important note: This review pertains only to ‘statistical aspects’ of the study and so ‘clinical aspects’ [like medical importance, relevance of the study, ‘clinical significance and implication(s)’ of the whole study, etc.] are to be evaluated [should be assessed] separately/independently. Further please note that any ‘statistical review’ is generally done under the assumption that (such) study specific methodological [as well as execution] issues are perfectly taken care of by the investigator(s). This review is not an exception to that and so does not cover clinical aspects {however, seldom comments are made only if those issues are intimately / scientifically related & intermingle with ‘statistical aspects’ of the study}. Agreed that ‘statistical methods’ are used as just tools here, however, they are vital part of methodology [and so should be given due importance].

COMMENTS: What exactly you want to convey when you say [in ‘Abstract-Methods] “matched control transplant recipients (TxCON, n=213), and healthy MVT participants (HCON, n=91)?”. My question is ‘how ‘matched control’(s) were selected [why ‘n’s so different, why (n=3–4 matched TxCON to n=1 transplant recipient participating in MVT) (line 173) then]? If you wish/want to indicate something other than usually known meaning of the term ‘matched control’, please explain. Otherwise explain ‘how done/achieved by method used’ [line 154: Members of the TxCON group were matched using propensity score matching on 4 parameters, need be explained though briefly].

Note that any regression techniques (of course, including general/ generalized linear mixed regression analysis) are not originally developed for comparison between the groups [please refer to lines 342-343: General(ized) linear regression analysis was also used to perform post-hoc testing for possible baseline differences in outcome variables between the study groups]. Also note that though the measures/tools used are appropriate, most of them {example: the 36-item Short Form Health Survey (SF-36), the Depression, Anxiety and Stress Scale (DASS-21), General Health Questionnaire (GHQ–12), etc., etc.} yield data that are in [at the most] ‘ordinal’ level of measurement [and not in ratio level of measurement for sure {as the score two times higher does not indicate presence of that parameter/phenomenon as double (for example, a Visual Analogue Scales VAS score or say ‘depression’ score)}]. Then application of suitable non-parametric test(s) is/are indicated/advisable [even if distribution may be ‘Gaussian’ (i.e. normal)]. Agreed that there is/are no non-parametric test(s)/technique(s) available to be used as alternative in all situation(s) [suitable / most desired/applicable], but should be used whenever/wherever they are available.

Considering aims of the study, I wonder, if inclusion of fourth group [sample of healthy MVT participants (HCON)] was needed at all? Are the “conclusion(s)” [particularly line 90 : Transplantoux's MVT exercise intervention barely affected PROs] given in ‘Abstract’ and given at end [lines 586 onwards] same/on similar lines? What exactly you want convey by the term ’barely affected’? Remember that “Absence of evidence is not evidence of absence” [Altman DG, Bland JM. BMJ volume 311, 1995, p 485 (Reprinted : Australian Veterinary Journal 1996;74, 311)]. {Even when P-value is not significantly lower (unfortunately there is no direct comparison between groups – which is desirable in this case) that is null hypothesis of no difference is not rejected, (in short, result is not significant), that does not amount to evidence of absence i.e. it does imply that there no difference. It only implies that there is no (i.e. these samples do not provide) enough evidence to prove (rather indicate with certain specified confidence level) the difference}.

I do not have any specific recommendation.

Reviewer #2: I would like to thank the authors for their time and effort in putting the manuscript together.

I think the most useful findings here are the barriers and facilitators data as there is limited research comparing to healthy controls and also comparing to more active transplant recipients. I think it shows that the least active recipients really do require more support and guidance in getting into exercise. I wondered if you might explore the type of transplant in the controls and see whether there are differences in variables? I would be interested to know if heart and lung transplant recipients have more favourable profiles given the more extensive rehabilitation. I think that the manuscript should be more dedicated to the barriers and facilitators work. I think the results speak for themselves that those who aspire to take part in the program are already motivated to exercise and perhaps want to 'push themselves' more by doing it, which explains the lack of effect that is seen.

6. PLOS authors have the option to publish the peer review history of their article (what does this mean?). If published, this will include your full peer review and any attached files.

Reviewer #1: No

Reviewer #2: No

---

## [Author Response · Author response to Decision Letter 0]

9 Jul 2022

Reviewer #1:

“Important note: This review pertains only to ‘statistical aspects’ of the study and so ‘clinical aspects’ [like medical importance, relevance of the study, ‘clinical significance and implication(s)’ of the whole study, etc.] are to be evaluated [should be assessed] separately/independently. Further please note that any ‘statistical review’ is generally done under the assumption that (such) study specific methodological [as well as execution] issues are perfectly taken care of by the investigator(s). This review is not an exception to that and so does not cover clinical aspects {however, seldom comments are made only if those issues are intimately / scientifically related & intermingle with ‘statistical aspects’ of the study}. Agreed that ‘statistical methods’ are used as just tools here, however, they are vital part of methodology [and so should be given due importance].”

We thank reviewer #1 for the expert comments on the statistical aspects of the present manuscript.

COMMENTS: 

Comment 1. “What exactly you want to convey when you say [in ‘Abstract-Methods] “matched control transplant recipients (TxCON, n=213), and healthy MVT participants (HCON, n=91)?”. My question is ‘how ‘matched control’(s) were selected [why ‘n’s so different, why (n=3–4 matched TxCON to n=1 transplant recipient participating in MVT) (line 173) then]? If you wish/want to indicate something other than usually known meaning of the term ‘matched control’, please explain. Otherwise explain ‘how done/achieved by method used’ [line 154: Members of the TxCON group were matched using propensity score matching on 4 parameters, need be explained though briefly].”

Thank you for this thoughtful comment. The original manuscript provided the following information on the selection method of control transplant recipients (line 169-172): “The TxCON group were selected from the UZ Leuven transplant database using propensity score matching based on the type of organ transplant, age, gender, and transplant vintage (n=3–4 matched TxCON to n=1 transplant recipient participating in MVT)”. 

To provide more detail regarding the methodology we used, we rephrased this paragraph in the revised manuscript (L171-179): “Patients from the TxCON group were selected from the UZ Leuven transplant database. Using uni- and multivariable logistic regression model analysis, a propensity score was estimated for all patients. Variables included in the propensity model were based on existing literature and included type of organ transplant, age, gender, and transplant vintage. Matching of these propensity scores did not result in an exact match for all parameters and therefore a multiple imputation was performed, resulting in different n (n=3–4 matched TxCON to n=1 transplant recipient participating in MVT). If more than 4 possible candidate TxCON patients were available, then a random selection of 4 was made.”

A larger TxCON group was chosen to optimise the use of the available variability in patient characteristics, thus optimizing stability and power of analyses. Having unequal group sizes is not an uncommon practice, for many possible reasons. In our case, the sizes of the intervention groups were constrained by the participants willing to participate. One obvious reason to match proportionally more controls to intervention subjects is that an oversupply of subject characteristics diminishes the chances of overfitting multiple regression analyses, and that –contrary to popular belief– larger control groups add statistical power to the study (Oldfield & Haig, https://philarchive.org/archive/OLDUSS).

Comment 2. Note that any regression techniques (of course, including general/ generalized linear mixed regression analysis) are not originally developed for comparison between the groups [please refer to lines 342-343: General(ized) linear regression analysis was also used to perform post-hoc testing for possible baseline differences in outcome variables between the study groups]. Also note that though the measures/tools used are appropriate, most of them {example: the 36-item Short Form Health Survey (SF-36), the Depression, Anxiety and Stress Scale (DASS-21), General Health Questionnaire (GHQ–12), etc., etc.} yield data that are in [at the most] ‘ordinal’ level of measurement [and not in ratio level of measurement for sure {as the score two times higher does not indicate presence of that parameter/phenomenon as double (for example, a Visual Analogue Scales VAS score or say ‘depression’ score)}]. Then application of suitable non-parametric test(s) is/are indicated/advisable [even if distribution may be ‘Gaussian’ (i.e. normal)]. Agreed that there is/are no non-parametric test(s)/technique(s) available to be used as alternative in all situation(s) [suitable / most desired/applicable], but should be used whenever/wherever they are available.

We thank reviewer #1 for these remarks. 

As mentioned by the reviewer, there are no alternative methods available for this data structure in the nonparametric techniques realm. It is true that using parametric techniques assume that distances are equal between values. However, while individual items might be ordinal, total scores of unidimensional scales are can be considered interval (Norman, 2010. Likert scales, levels of measurement and the "laws" of statistics). There is ample evidence that parametric techniques are at least as robust for ordinal/interval data, provided that they fit the data. This implies that the distribution needs to be symmetrical (p-values are affected by long-tailed skewness) and that possible nonlinearities need to be checked and, if needed, modeled (which ranking procedures handle automatically). In such cases, one can reliably conclude that a change of one unit in the independent variable is associated with an ‘x’ change in the dependent, subjective value. In the end, all modeling portrays a representation of reality but is never exact.

With regard to the generalized modeling (as an ordinal regression analysis), the total score needed to be ordinally categorized, as algebraic transformation to a normal distribution was not possible. Here too, several checks are required for the parametric model to fit. First, the proportionality of the odds ratios across the different cut offs needs to hold more or less (the alternative name for this procedure is a ‘proportional odds model’). Second, the cutoffs need to be varied to check whether the result is stable with regard to the categories chosen. Such sensitivity analyses, which are part of any standard model validation, to ensure that the model describes the data as well as possible.

That regression techniques are not developed for group comparison does not need to be a drawback. After a century of tweaking testing procedures, the fact that models are called ‘general’ or ‘generalized’ implies that the so called t-Tests, ANOVAs, ANCOVAs, and logistic or mixed regressions are all special cases of a more overarching approach. It is for instance perfectly possible to perform a t-Test using a generalized mixed regression model if the distribution requirement of the response variable is set at ‘normal’, random-effects variance at zero, and the predictor interpreted as a binary variable or a contrast between its two values; overkill if it needs to be calculated by hand, but the computer does the calculations and gives the same result as with the more ‘specialized’ t-Test. The same equivalence exists between chi-square and logistic/multinomial regression techniques. Ordinal regression is indeed a bit different from the traditional ordinal techniques, however, if done well, provides a useful alternative.

Comment 3. Considering aims of the study, I wonder, if inclusion of fourth group [sample of healthy MVT participants (HCON)] was needed at all? Are the “conclusion(s)” [particularly line 90 : Transplantoux's MVT exercise intervention barely affected PROs] given in ‘Abstract’ and given at end [lines 586 onwards] same/on similar lines? What exactly you want convey by the term ’barely affected’? Remember that “Absence of evidence is not evidence of absence” [Altman DG, Bland JM. BMJ volume 311, 1995, p 485 (Reprinted : Australian Veterinary Journal 1996;74, 311)]. {Even when P-value is not significantly lower (unfortunately there is no direct comparison between groups – which is desirable in this case) that is null hypothesis of no difference is not rejected, (in short, result is not significant), that does not amount to evidence of absence i.e. it does imply that there no difference. It only implies that there is no (i.e. these samples do not provide) enough evidence to prove (rather indicate with certain specified confidence level) the difference}.

Thank you for these questions. 

In our opinion, the inclusion of HCON provides a much-needed reference against which patient-reported outcomes from TxCON, TxCYC, and TxHIK can be contrasted and interpreted. Thanks to the inclusion of HCON, it became clear that transplant patients participating in the MVT program (TxCYC and TxHIK) represent a selected group of patients with very similar patient-reported outcomes compared to healthy study participants (HCON), but with a distinct profile compared to control transplant patients (TxCON). This is in fact a unique comparison adding value to the field. As reviewer #2 stated: “…the most useful findings here are the barriers and facilitators data as there is limited research comparing to healthy controls...”.

At least to our understanding, the conclusions from the abstract and from the main text are indeed the same: a vast array of patient-reported outcomes assessed throughout the study did not respond to the MVT exercise intervention. The word ‘barely’ thus refers to the fact that almost none of the patient-reported outcomes changed due to the exercise intervention (referring to what was stated in the result section of the abstract). In attempt to avoid any misunderstanding, we rephrased this sentence in the abstract to: “Barely any of the PROs assessed in the present study responded to Transplantoux's MVT exercise intervention.”

I do not have any specific recommendation.

We thank expert reviewer #1 again for the interesting remarks and hope our responses and revisions meet the standards of the reviewer.

 

Reviewer #2: 

I would like to thank the authors for their time and effort in putting the manuscript together.

I think the most useful findings here are the barriers and facilitators data as there is limited research comparing to healthy controls and also comparing to more active transplant recipients. I think it shows that the least active recipients really do require more support and guidance in getting into exercise. I wondered if you might explore the type of transplant in the controls and see whether there are differences in variables? I would be interested to know if heart and lung transplant recipients have more favourable profiles given the more extensive rehabilitation. I think that the manuscript should be more dedicated to the barriers and facilitators work. I think the results speak for themselves that those who aspire to take part in the program are already motivated to exercise and perhaps want to 'push themselves' more by doing it, which explains the lack of effect that is seen.

We thank reviewer #2 for these thoughtful comments and suggestions. We addressed each comment/question separately below.

Comment 1. “I think the most useful findings here are the barriers and facilitators data as there is limited research comparing to healthy controls and also comparing to more active transplant recipients. I think it shows that the least active recipients really do require more support and guidance in getting into exercise.”

We agree with reviewer #2 that, at least in part due to the absence of major intervention effects on PROs, the barriers and facilitators data are among the most interesting findings of the present study.

We thank reviewer #2 for acknowledging that the present study covers a research gap regarding the comparison of control transplant recipients, a subpopulation of active transplant recipients, and healthy controls.

We fully agree with reviewer #2 that the present study indicates that the least active transplant recipients require the most support and guidance in becoming more physical active and/or engaging in physical exercise. This is an important message, and points to a need for increasing efforts to ‘reach’ the least physically active who can benefit most from more physical activity. In order to achieve this ‘reach’, our findings on specific ‘barriers’ for this specific patient group provide an important starting point for targeted intervention development as well as successful implementation. We now further highlighted these messages in the manuscript:

- First paragraph of the Discussion, line 462-468: “Lastly, the present study identified specific barriers and motivators to physical activity in TxCON as compared to TxCYC, TxHIK, and HCON. These insights provide a crucial building block for intervention development and choice of relevant implementation strategies for these least active transplant recipients (TxCON) that could benefit most from getting more physically active. This area of research should get priority as currently these patients typically are not reached in RCTs nor in daily clinical practice.”

- Discussion line 531-545: “Specific motivators and barriers identified in TxCON can support current physical activity initiatives, such as for instance organized by local patient organizations (e.g., Transplantoux) or international federations (e.g., World Transplant Games Federation) to more successfully target less or non-physically active patients. Especially developing adapted interventions and use of specific implementation strategies can support to reach these patients and to get and keep them more physically active. RCTs show a bias towards inclusion of more physically active patients or patients willing to engage in physical activity [62,63]. Also in Transplantoux this has been the case, resulting in a selection bias. Sampling strategies should guarantee that the whole spectrum of transplant patients, ranging from not physically active at all to already physically active, are included in intervention studies. An approach tailored to the needs and preferences of the physically inactive transplant recipients should focus on finding strategies first to activate these patients, then to keep them physically active. Regarding this point, our TxCON’s perceived barriers and facilitators offer a useful starting point [64,65].” The following references were added: 

o [62]: De Smet S, Van Craenenbroeck AH. Exercise training in patients after kidney transplantation. Clin Kidney J. 2021;14(Suppl 2):ii15–ii24 

o [63]: De Smet S, O’Donoghue K, Lormans M, Monbaliu D, Pengel L. Does exercise training improve physical fitness and health in adult liver transplant recipients? A systematic review and meta-analysis. Transplantation. 2022;accepted for publication. 

o [65]: Leunis S, Vandecruys M, Cornelissen V, Van Craenenbroeck AH, De Geest S, Monbaliu D, et al. Physical activity behaviour in solid organ transplant recipients: proposal of theory‐driven physical activity interventions. Kidney Dial. 2022;2(2):298–329.

Comment 2. “I wondered if you might explore the type of transplant in the controls and see whether there are differences in variables? I would be interested to know if heart and lung transplant recipients have more favourable profiles given the more extensive rehabilitation.”

Upon the thoughtful request of the reviewer, we explored the role of the transplant type in TxCON regarding MET-minutes per week, without finding any differences. Further differences with other variables were not explored, since that would add an extra study aim (worthy of a separate paper). In case there would be differences in the variables that now remain unexplored, that could be a reason to control for organ group in the analysis. However, by matching on transplanted organ, we already implicitly controlled for any possible confounding. Hence, the danger of overlooking any bias potential in this regard remains negligible.

Comment 3. “I think that the manuscript should be more dedicated to the barriers and facilitators work.”

In order to meet the suggestion of reviewer #2, we have now further emphasized the presence and implications of the identified barriers and facilitators profiles (cf. comment 1):

- First paragraph of the Discussion, line 462-468: “Lastly, the present study identified specific barriers and motivators to physical activity in TxCON as compared to TxCYC, TxHIK, and HCON. These insights provide a crucial building block for intervention development and choice of relevant implementation strategies for these least active transplant recipients (TxCON) that could benefit most from getting more physically active. This area of research should get priority as currently these patients typically are not reached in RCTs nor in daily clinical practice.”

- Discussion line 531-545: “Specific motivators and barriers identified in TxCON can support current physical activity initiatives, such as for instance organized by local patient organizations (e.g., Transplantoux) or international federations (e.g., World Transplant Games Federation) to more successfully target less or non-physically active patients. Especially developing adapted interventions and use of specific implementation strategies can support to reach these patients and to get and keep them more physically active. RCTs show a bias towards inclusion of more physically active patients or patients willing to engage in physical activity [62,63]. Also in Transplantoux this has been the case, resulting in a selection bias. Sampling strategies should guarantee that the whole spectrum of transplant patients, ranging from not physically active at all to already physically active, are included in intervention studies. An approach tailored to the needs and preferences of the physically inactive transplant recipients should focus on finding strategies first to activate these patients, then to keep them physically active. Regarding this point, our TxCON’s perceived barriers and facilitators offer a useful starting point [64,65].”

Comment 4. “I think the results speak for themselves that those who aspire to take part in the program are already motivated to exercise and perhaps want to 'push themselves' more by doing it, which explains the lack of effect that is seen.”

We agree with the interpretation of reviewer #2. Furthermore, it is likely that physical activity intervention effects would be more pronounced in physically inactive transplant recipients, but such hypothesis remains to be confirmed in future studies.

---

## [Decision Letter · Decision Letter 1]

10 Aug 2022

Patient-reported outcomes evaluation of the MVT exercise intervention and assessment of facilitators and barriers to physical activity in transplant recipients

PONE-D-21-28527R1

Dear Dr. De Geest,

We’re pleased to inform you that your manuscript has been judged scientifically suitable for publication and will be formally accepted for publication once it meets all outstanding technical requirements.

Kind regards,

James Mockridge

Staff Editor

PLOS ONE

Reviewers' comments:

Reviewer's Responses to Questions

**Comments to the Author**

1. If the authors have adequately addressed your comments raised in a previous round of review and you feel that this manuscript is now acceptable for publication, you may indicate that here to bypass the “Comments to the Author” section, enter your conflict of interest statement in the “Confidential to Editor” section, and submit your "Accept" recommendation.

Reviewer #1: (No Response)

Reviewer #2: All comments have been addressed

2. Is the manuscript technically sound, and do the data support the conclusions?

Reviewer #1: (No Response)

Reviewer #2: Yes

3. Has the statistical analysis been performed appropriately and rigorously? 

Reviewer #1: (No Response)

Reviewer #2: Yes

4. Have the authors made all data underlying the findings in their manuscript fully available?

Reviewer #1: (No Response)

Reviewer #2: Yes

5. Is the manuscript presented in an intelligible fashion and written in standard English?

Reviewer #1: (No Response)

Reviewer #2: Yes

6. Review Comments to the Author

Reviewer #1: COMMENTS: Since all of the comments made on earlier draft were/are answered [though not attended/followed and I am not very much convinced for reasons given or arguments made, and so not very happy], I recommend the acceptance though, in my opinion the argued points are not always true.

Reviewer #2: I would like to thank the authors for addressing my comments and making some thorough changes to the manuscript.

7. PLOS authors have the option to publish the peer review history of their article (what does this mean?). If published, this will include your full peer review and any attached files.

Reviewer #1: No

Reviewer #2: No

---

## [Editor Report · Acceptance letter]

18 Oct 2022

PONE-D-21-28527R1 

Patient-reported outcomes evaluation and assessment of facilitators and barriers to physical activity in the Transplantoux aerobic exercise intervention 

Dear Dr. De Geest:

I'm pleased to inform you that your manuscript has been deemed suitable for publication in PLOS ONE. Congratulations! Your manuscript is now with our production department. 

Kind regards, 

on behalf of

Dr James Mockridge 

Staff Editor

PLOS ONE